# I2I-Galip: Unsupervised Medical Image Translation Using Generative Adversarial CLIP

**Yilmaz Korkmaz**[1]                                                    YKORKMA1@JHU.EDU
**Vishal M. Patel**[1]                                                   VPATEL36@JHU.EDU
[1] *Johns Hopkins University*

**Editors:** Accepted for publication at MIDL 2025

## Abstract

Unpaired image-to-image translation is a challenging task due to the absence of paired examples, which complicates learning the complex mappings between the distinct distributions of the source and target domains. One of the most commonly used approaches for this task is cycle-consistent models which require the training of a new pair of generator-discriminator networks for each translation. In this paper, we propose a new image-to-image translation framework named Image-to-Image-Generative-Adversarial-CLIP (I2I-Galip) where we utilize pre-trained multi-modal foundation models to mitigate the need of separate generator-discriminator pairs for each source-target mapping while achieving better and more efficient multi-domain translation. By utilizing the massive knowledge gathered during pre-training a foundation model, our approach makes use of a single lightweight generator network with $\approx$13M parameters for the multi-domain image translation task. Comprehensive experiments on translation performance in public MRI and CT datasets show the superior performance of the proposed framework over the existing approaches.

## 1. Introduction

Medical image translation is a crucial task due to the availability of diverse information across various modalities. However, it is challenging because of significantly different domain distributions, necessitating the learning of very complex mappings between different imaging modalities (Roy et al., 2013). Many supervised deep learning-based image translation methods have been proposed to address this problem (Dar et al., 2019; Jiang et al., 2023; Armanious et al., 2020). However, these methods are limited due to the requirement of paired training data which might be challenging to acquire in real case scenarios. To overcome this constraint, various unsupervised image translation methods have been introduced for both general computer vision and medical imaging tasks (Dai et al., 2020; Liu et al., 2017; Huang et al., 2018; Özbey et al., 2023; Han et al., 2021; Yi et al., 2017; Torbunov et al., 2023). CycleGAN (Zhu et al., 2017) is one of the first approaches that proposed unpaired image translation which loosened the requirement for paired datasets by enforcing cycle-consistency among inverse translations. However, in the case of multiple modalities, cycle-consistent models introduce significant computational requirements as separate generator-discriminator pairs are required for each new modality. To mitigate the need of separate network pairs several multi-domain translation frameworks have been proposed (Choi et al., 2018, 2020; Huang et al., 2018; Lee et al., 2018). Nonetheless, these methods generally lag in performance compared to uni-modal approaches.

More recently, a couple of text-driven diffusion based image-to-image translation frameworks have been proposed that integrate large vision-language pre-trained models as guidance (Tumanyan et al., 2023; Rombach et al., 2022; Hertz et al., 2022; Kwon and Ye, 2022), enabling robust translation across multiple domains. While these models provide zero-shot editing capabilities for various text conditions, they are limited in delivering fidelity necessary for the medical tasks. Moreover, these methods impose a significant computational burden due to the requirement for large denoiser backbones and extended inference times in their backward diffusion processes.

In this paper, we propose a cycle-consistent generative adversarial model to address the aforementioned limitations. Our model integrates BiomedCLIP (see Section 2.2), a pre-trained multi-modal vision-language model specifically trained in the medical domain, within a cycle-consistent feed-forward framework. By leveraging contrastive information from this large pre-trained network, we eliminate the need to train a new generator network for each translation task and reduce the requirement for large discriminator backbones in feature extraction. Furthermore, our model enhances overall translation performance compared to existing unsupervised approaches in both single and multi-domain translation tasks.

Our main contributions can be summarized as follows:

- We introduce a novel adversarial framework for language-driven multi-domain medical image translation.

- Our framework outperforms existing unsupervised baselines with a relatively lightweight backbone. Extensive experiments demonstrate its superior performance across various publicly available datasets from different modalities.

## 2. Background

### 2.1. Cycle-Consistent Generative Adversarial Networks (CycleGAN)

CycleGAN (Zhu et al., 2017) models the unpaired image translation problem between domain $A$ and $B$ using two translators. First, two translators $(G : A \to B)$ and $(F : B \to A)$ are defined. Then G and F are forced to be inverses of each other, thus making both mappings to be approximately bijections. CycleGAN achieves remarkable performance using this cycle-consistency combined with the adversarial loss which encourages $F(G(X_A)) \approx X_A$ and $G(F(X_B)) \approx X_B$.

### 2.2. BiomedCLIP

In this paper, we utilize BiomedCLIP (Zhang et al., 2023) as our pre-trained vision-language model. BiomedCLIP is trained on PMC-15M dataset using pairs of figures and captions from biomedical research articles in PubMed Central and outperforms other medical vision-language models in various tasks (Zhang et al., 2023). BiomedCLIP utilizes a ViT-B (Dosovitskiy et al., 2020) based image encoder to generate image embeddings while utilizing PubMedBERT (Gu et al., 2021) for the text embeddings.

## 3. Methodology

### 3.1. I2I-Galip

We design a lightweight generator network which is a very thin variant of the latent diffusion U-Net (Rombach et al., 2022) (with only $\approx$ 13M parameters). Our discriminator network uses the projections of intermediate Vision Transformer (ViT) features as input, adapted from text-to-image model Stylegan-T (Sauer et al., 2023). This discriminator design allow us to utilize the output of different layers in BiomedCLIP's ViT, capturing different level of details. We modify this design by dividing the discriminator heads into distinct sets, tailored specifically for a target translation domain. We also utilize BiomedCLIP's text encoder to generate target text embeddings using captions for each modality, which controls the generated image features via cross-attention transformers while serving as a regularizer in the training (see Figure 1a). Overall training objective for the generator can be expressed as follows

$$
\begin{aligned}
\mathcal{L}_{total} =& \lambda_{cycle} \cdot \mathcal{L}_{cycle} + \lambda_{adv} \cdot \mathcal{L}_{adv_G} + \lambda_{clip} \cdot \mathcal{L}_{clip} \\
&+ \lambda_{cls} \cdot \mathcal{L}_{cls} + \lambda_{identity} \cdot \mathcal{L}_{identity},
\end{aligned}
\tag{1}
$$

where $\lambda_{cycle}, \lambda_{adv}, \lambda_{clip}, \lambda_{cls}, \lambda_{identity}$ are coefficients to control the contribution from each loss. We denote the loss associated with the discriminator as $\mathcal{L}_{adv_D}$. In what follows, we describe each of these loss terms in detail.

1. **Adversarial Loss**: By leveraging intermediate features from the ViT, direct feature extraction from images becomes unnecessary, enabling the use of lightweight discriminator heads for each feature level. We utilize the least squares GAN loss (Mao et al., 2017) to enhance the stability of training instead of Hinge loss used in StyleGAN-T, which can be defined as follows

$$
\begin{aligned}
\mathcal{L}_{adv_G} =& \mathbb{E}[(Head_A(E_{X_A}^{out}, E_{T_A}) - 1)^2] \\
&+ \mathbb{E}[(Head_B(E_{X_B}^{out}, E_{T_B}) - 1)^2],
\end{aligned}
\tag{2}
$$

$$
\begin{aligned}
\mathcal{L}_{adv_D} =& \mathbb{E}\left[(Head_A(E_{X_A}^{input}, E_{T_A}) - 1)^2 + (Head_A(E_{X_A}^{out}, E_{T_A}))^2\right] \\
&+ \mathbb{E}\left[(Head_B(E_{X_B}^{input}, E_{T_B}) - 1)^2 + (Head_B(E_{X_B}^{out}, E_{T_B}))^2\right],
\end{aligned}
\tag{3}
$$

where $Head_A$ and $Head_B$ correspond to the discriminator heads allocated for the specific target domain, $E_{T_A}$ and $E_{T_B}$ are corresponding text encodings (i.e., text encodings of captions for each domain), $E_{X_A}^{input}$, $E_{X_B}^{input}$, $E_{X_A}^{out}$ and $E_{X_B}^{out}$ are feature maps from ViT for input and generated images from domain $A$ and $B$, respectively.

2. **Cycle Loss**: We enforce cycle-consistency loss (Zhu et al., 2017), shown in Figure 1b, to enforce faithful translation between source and target domains for each pair:

$$
\begin{aligned}
\mathcal{L}_{\text{cycle}} =& \mathbb{E}\left[\|X_B^{input} - G(X_A^{input}, E_{T_B}, Y_B)\|_1\right] \\
&+ \mathbb{E}\left[\|X_A^{input} - G(X_B^{input}, E_{T_A}, Y_A)\|_1\right].
\end{aligned}
\tag{4}
$$

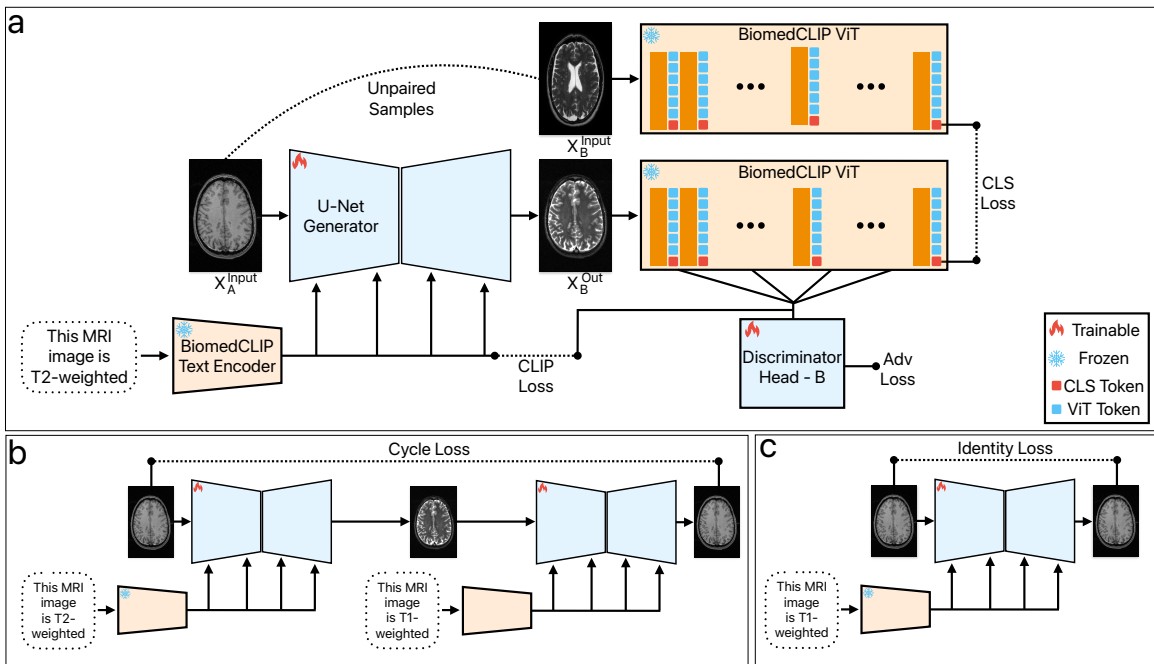

**Figure 1:** Training scheme and overall model architecture of I2I-Galip is illustrated for the $T_1$- to $T_2$-weighted MRI translation task. Part (a) illustrates the definition of $L_{clip}$, $L_{cls}$ and $L_{adv}$ losses. $X_A^{Input}$, $X_B^{Input}$ and $X_B^{Out}$ denotes the input $T_1$-weighted, input $T_2$-weighted and output $T_2$-weighted images respectively. Part (b) and (c) stand for $L_{cycle}$ and $L_{identity}$ losses respectively. BiomedCLIP's ViT and Text Encoder parameters are frozen during training. "This MRI Image is $T_2$-weighted" corresponds to a sample prompt used for $T_1$ to $T_2$ translation.

3. **CLIP Loss**: We minimize the cosine distance between the text encoding corresponds to the caption of target domain (e.g., "This MRI image is $T_1$-Weighted") and the encoding from ViT for the generated images to enable the utilization of CLIP's joint embedding space similarly with (Patashnik et al., 2021), which can be defined as follows

$$\mathcal{L}_{\text{clip}} = -\frac{\langle E_{T_A}, E_{X_A}^{out^{last}}\rangle}{\|E_{T_A}\| \cdot \|E_{X_A}^{out^{last}}\|} - \frac{\langle E_{T_B}, E_{X_B}^{out^{last}}\rangle}{\|E_{T_B}\| \cdot \|E_{X_B}^{out^{last}}\|}, \tag{5}$$

where $E_{X_A}^{out^{last}}$, $E_{X_B}^{out^{last}}$ are the image encodings from the last layer of ViT for the generated images from domain $A$ and $B$, respectively. Generally, $\mathcal{L}_{clip}$ is dominated by cycle and adversarial losses giving comparingly small benefits (see Section 4.1 for details).

4. **CLS Loss**: The CLS tokens in the final layers of the vision transformers are recognized for containing semantically rich information, as highlighted by (Tumanyan et al., 2022), which is typically leveraged for downstream classification tasks and shown to be beneficial in image translation (Kwon and Ye, 2022). Therefore, we enforce the cosine similarity between the CLS tokens in the ViT for the generated and target

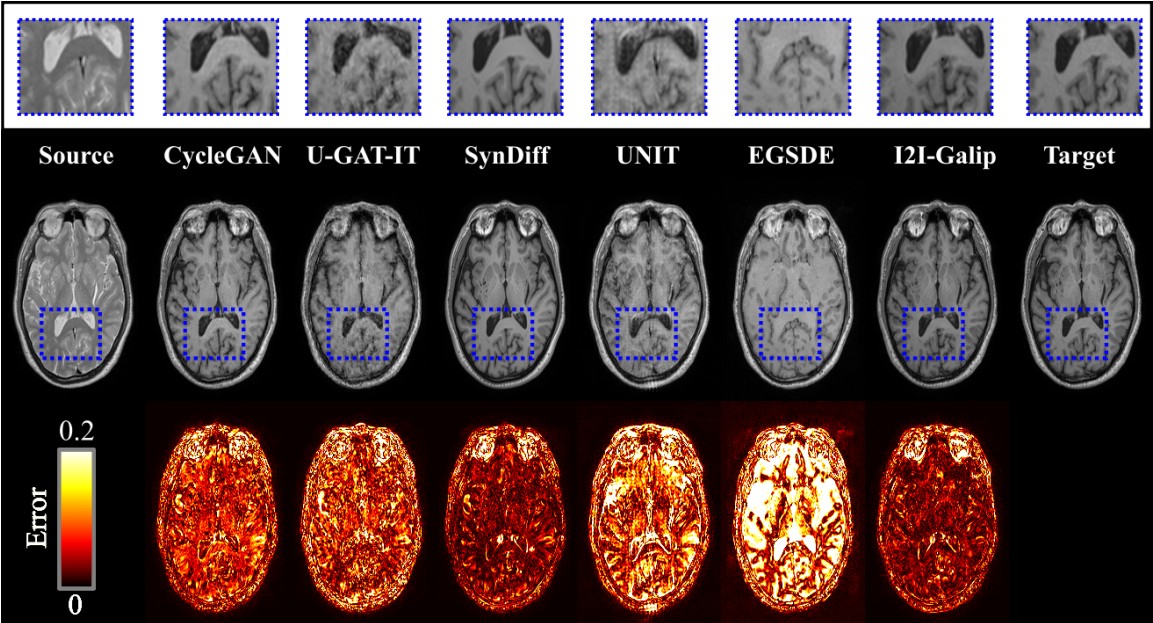

**Figure 2:** Multi-domain translation illustrations from PD to $T_1$-weighted image in IXI dataset. Accompanying this are error maps and magnified sections, positioned below and above each translation, respectively.

domain's images to enforce semantic similarity among these images, which can be written as follows

$$\mathcal{L}_{\text{cls}} = -\frac{\langle cls_{X_A}^{input}, cls_{X_A}^{out}\rangle}{\|cls_{X_A}^{input}\| \cdot \|cls_{X_A}^{out}\|} - \frac{\langle cls_{X_B}^{input}, cls_{X_B}^{out}\rangle}{\|cls_{X_B}^{input}\| \cdot \|cls_{X_B}^{out}\|}, \tag{6}$$

where $cls_{X_A}^{input}$ and $cls_{X_B}^{input}$ are the CLS tokens in the last layers of ViT for input images from domain $A$ and $B$ respectively. $cls_{X_A}^{out}$ and $cls_{X_B}^{out}$ are corresponding CLS tokens for the generated images.

5. **Identity Loss**: The identity loss is found to be beneficial to maintain source image structure in translation by enforcing the pixel-level equality when target and source domains match (Zhu et al., 2017). We enforce it via using same labels and text embeddings corresponding to the input image domain (see Figure 1c).

### 3.2. Datasets

We conduct experiments on single-coil brain MRI dataset (IXI) and CT-MRI dataset (Nyholm et al., 2018) to demonstrate the performance of our approach. Dataset details are presented in the Section B.1. We consider the IXI dataset in both multi-domain and single-domain translation contexts. In the multi-domain scenario, we use a single network for all translation tasks, whereas in the single-domain scenario, we utilize distinct networks for each individual task. On the other hand, CT-MRI dataset only allows us to use single-domain translation context.

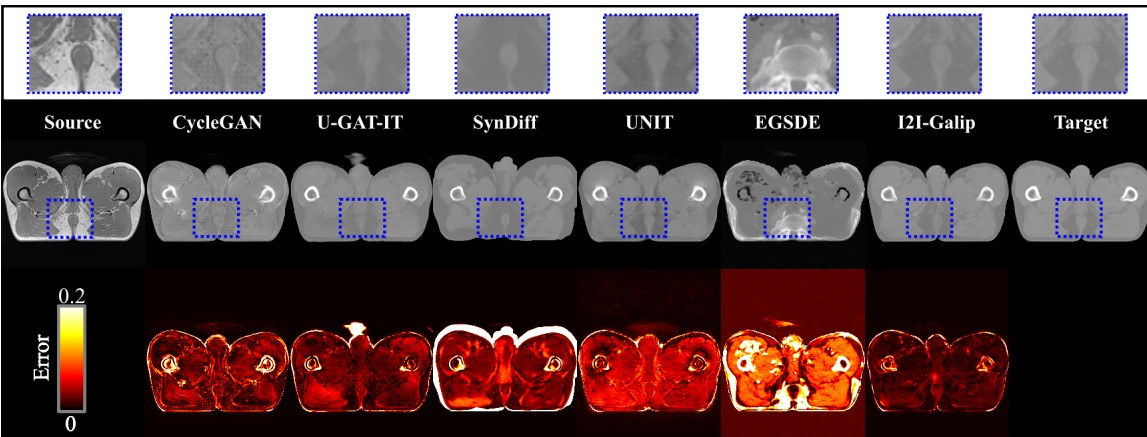

**Figure 3:** Single-domain translation from $T_1$-weighted Pelvic MRI to CT images. Accompanying this are error maps and magnified sections, positioned below and above each translation, respectively.

**Table 1:** Multi-domain image translation results in IXI dataset. $T_1$-, $T_2$- and PD-weighted images are considered. Best and second best results are indicated with red and blue respectively.

| One-to-one task | $T_1$->$T_2$ | $T_2$->$T_1$ | $T_2$->PD | PD->$T_2$ | $T_1$->PD | PD->$T_1$ |
|---|---|---|---|---|---|---|
| IXI | PSNR \| SSIM | PSNR \| SSIM | PSNR \| SSIM | PSNR \| SSIM | PSNR \| SSIM | PSNR \| SSIM |
| I2I-Galip-M | 27.22 \| 90.18 | 27.30 \| 90.86 | 32.34 \| 95.74 | 33.12 \| 95.39 | 26.76 \| 90.75 | 27.70 \| 91.20 |
| I2I-Galip-S | 27.47 \| 90.54 | 27.33 \| 91.06 | 32.11 \| 95.65 | 32.87 \| 95.62 | 26.99 \| 90.80 | 27.75 \| 91.07 |
| CycleGAN | 26.10 \| 87.36 | 26.31 \| 88.51 | 27.43 \| 93.68 | 31.07 \| 93.81 | 24.56 \| 88.26 | 25.91 \| 89.47 |
| U-GAT-IT | 24.44 \| 86.19 | 24.51 \| 86.85 | 26.81 \| 91.39 | 29.03 \| 92.11 | 22.98 \| 85.16 | 24.83 \| 87.44 |
| SynDiff | 26.34 \| 91.87 | 27.60 \| 92.14 | 33.15 \| 96.87 | 29.81 \| 96.99 | 27.29 \| 92.49 | 25.54 \| 92.41 |
| UNIT | 23.59 \| 84.40 | 24.76 \| 86.63 | 25.22 \| 91.42 | 29.10 \| 93.30 | 23.20 \| 86.00 | 23.50 \| 80.05 |
| EGSDE | 16.93 \| 53.32 | 17.44 \| 57.54 | 17.98 \| 75.93 | 16.40 \| 57.55 | 19.70 \| 71.21 | 19.71 \| 59.73 |

### 3.3. Implementation Details

We illustrate the model complexities using the number of parameters in each competing method in the Table 3. A single NVIDIA RTX A5000 GPU with PyTorch framework is utilized in all experiments. Our model is trained with Adam optimizer with an initial learning rate set at 0.0002, which is linearly decreased to 0 after the $50th$ epoch. Number of discriminator head sets are determined according to the number of domains in the translation problem, where for IXI it is 3, and 2 for CT-MRI. We utilize hyperparameters 10, 1, 1, 1, 1 for $\lambda_{cycle}$, $\lambda_{adv}$, $\lambda_{cls}$, $\lambda_{clip}$, and $\lambda_{identity}$ respectively.

### 4. Results

We utilize well known unsupervised image translation baselines CycleGAN (Zhu et al., 2017), U-GAT-IT (Kim et al., 2019), SynDiff (Özbey et al., 2023), UNIT (Liu et al., 2017) and EGSDE (Zhao et al., 2022) as competing methods (see Section B.2 for details). We use Peak-Signal-to-Noise-Ratio (PSNR, dB) and Structural Similarity Index Measure (SSIM, %) to compare the translation performances of competing methods. Results are presented

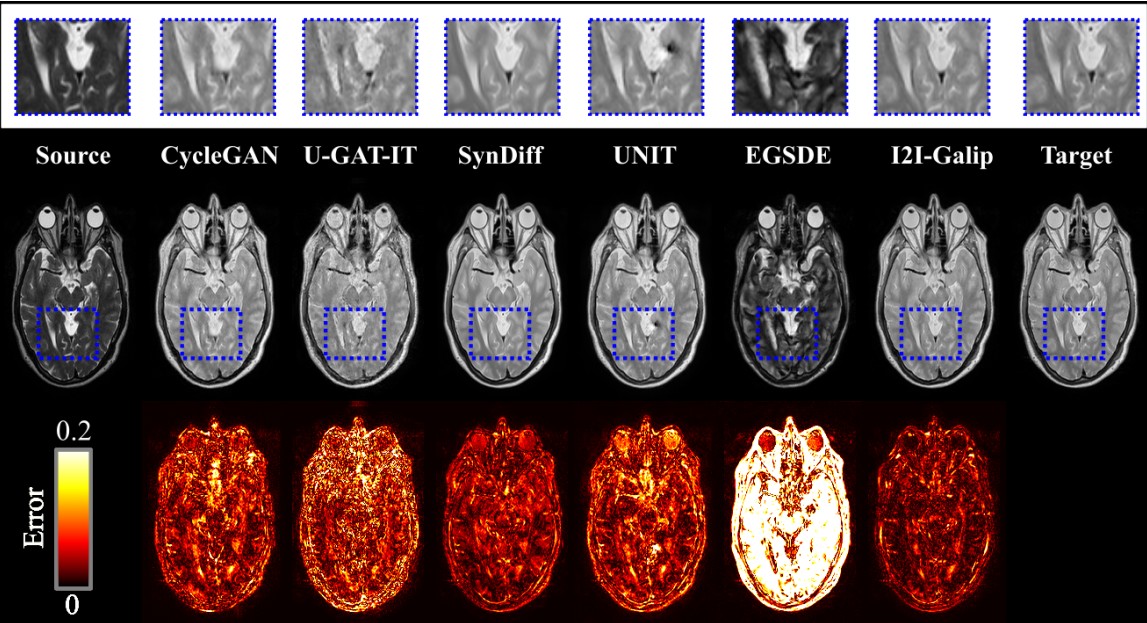

**Figure 4:** Multi-domain translation illustrations from $T_2$-weighted to PD image in IXI dataset. Accompanying this are error maps and magnified sections, positioned below and above each translation, respectively.

for both single- and multi-domain case in IXI for I2I-Galip to show the effectiveness of the proposed approach for both cases. CT-MRI results are presented as the single-domain translation. I2I-Galip-S (Single), CycleGAN, U-GAT-IT, UNIT and SynDiff are separately trained for all possible domain pairs while I2I-Galip-M (Multi) is trained once per dataset. EGSDE is an unsupervised image translation method that is agnostic to translation direction. However, it relies on separately pre-trained diffusion models for each target domain. Table 1 and Table 2 show the translation performance in IXI and CT-MRI datasets, respectively. We show the corresponding translated images for each competing methods for distinct translation tasks in Figure 2, Figure 3, and Figure 4. Best and second best performances are highlighted as red and blue, respectively, in each table for each metric.

Overall, I2I-Galip-M, a single network for multi-domain translation in IXI—unlike CycleGAN, which requires separately trained models for each image pair—achieves 2.17dB higher PSNR and over 2% better SSIM compared to CycleGAN, the foundational approach. Additionally, I2I-Galip-S outperforms the second-best method with a 0.10 dB improvement in PSNR and a 1.52% increase in SSIM for the $T_1$ to CT translation, also delivers a 1.38% gain in SSIM for the $T_2$ to CT task. Compared to SynDiff—a state-of-the-art diffusion-based, cycle-consistent translation model—I2I-Galip achieves comparable performance while significantly reducing computational demands. For example, in the $T_1$ to $T_2$ task, I2I-Galip-M attains 27.22 dB PSNR (compared to SynDiff's 26.34 dB), and in the PD to $T_2$ task, it reaches 33.12 dB PSNR versus SynDiff's 29.81 dB. Similar performance trends are observed across other tasks, demonstrating that I2I-Galip matches SynDiff's effectiveness while preserving finer structural details, avoiding the oversmoothing effect seen

in SynDiff, as illustrated in the qualitative figures. Additionally, I2I-Galip requires far fewer network parameters (see Table 3) and offers faster inference by circumventing the iterative diffusion process inherent to SynDiff (see Table 4 for details). Unlike other competing methods such as CycleGAN, U-GAT-IT, and UNIT, which suffer from noise artifacts that degrade output quality, I2I-Galip also effectively avoids these issues while maintaining sharp, accurate translations.

**Table 2:** Single-domain image translation results in CT-MRI dataset for $T_1$- and $T_2$-weighted images.

|  | $T_1$->CT | | $T_2$->CT | |
|---|---|---|---|---|
|  | PSNR | SSIM | PSNR | SSIM |
| I2I-Galip | 26.13 | 90.86 | 27.08 | 91.30 |
| CycleGAN | 24.55 | 78.63 | 27.39 | 89.92 |
| U-GAT-IT | 25.79 | 89.34 | 26.01 | 87.48 |
| SynDiff | 23.81 | 75.16 | 21.73 | 75.07 |
| UNIT | 26.02 | 79.22 | 25.15 | 75.30 |
| EGSDE | 19.03 | 74.63 | 14.74 | 66.67 |

**Table 3:** Model complexities are illustrated using total number of parameters for each competing method. The third row indicates the number of required generator and discriminator networks, given the specified number of domain. $T$ and P(.) represents the number of domains for a multi-modal translation problem and permutation operator respectively. Total parameters are calculated for a representative case where $T = 4$.

| Network/Model | I2I-Galip | CycleGAN | U-GAT-IT | SynDiff | UNIT | EGSDE |
|---|---|---|---|---|---|---|
| Generator (G) | 13.2M | 11.3M | 278.9M | 39.7M + 7.8M | 5.4M + 5.4M | 164M |
| Discriminator (D) | 23.9M | 2.7M | 56.4M | 27.7M + 2.7M | 2.8M | 0 |
| Times (G, D) | 1, T | P(T,2), P(T,2) | P(T,2), P(T,2) | P(T,2), P(T,2) | P(T,2), P(T,2) | T, 0 |
| Total | 108.8M | 169.5M | 4023.6M | 936.6M | 162.2M | 657.2M |

**Table 4:** Memory usage, training time, and inference time for the most lightweight and the most computationally intensive methods for a single-domain translation with a NVIDIA RTX A5000 gpu.

|  | CycleGAN | I2I-Galip | SynDiff |
|---|---|---|---|
| Memory | 3,128 MiB | 7,074 MiB | 9,638 MiB |
| Training Time | ≈4.5h | ≈27h | ≈35h |
| Inference Time | 0.00353s | 0.04883s | 0.1792s |

### 4.1. Ablation Studies

As shown in Table 5, we assess the contribution of each loss component and compare I2I-Galip with BiomedCLIP against its variant equipped with OpenCLIP (Ilharco et al., 2021) trained on Liaon (Schuhmann et al., 2022) in both single- and multi-domain settings. The majority of the performance gains stem from adversarial and cycle losses, although CLIP, identity, and CLS losses add notable benefits in the multi-domain scenario. In contrast, the

adversarial loss tends to dominate in the single-domain case, reducing the impact of the other loss terms. We discuss the underlying reasons for these findings in Section 5.

**Table 5:** Single- and multi-domain ablation results in IXI dataset. PSNR and SSIM values are averaged across the whole test set.

|  | I2I-Galip-S | | I2I-Galip-M | |
|---|---|---|---|---|
|  | IXI | | IXI | |
|  | PSNR | SSIM | PSNR | SSIM |
| Proposed | 29.09 | 92.48 | 29.07 | 92.35 |
| $\lambda_{adv} = 0$ | 19.80 | 60.80 | 18.62 | 49.05 |
| $\lambda_{cls} = 0$ | 29.00 | 92.26 | 28.88 | 91.99 |
| $\lambda_{cycle} = 0$ | 27.91 | 90.93 | 28.08 | 90.88 |
| $\lambda_{clip} = 0$ | 28.90 | 92.23 | 28.99 | 92.18 |
| $\lambda_{identity} = 0$ | 28.74 | 91.12 | 28.76 | 92.01 |
| CLIP-Laion-2B | 19.06 | 66.67 | 27.54 | 91.21 |

## 5. Discussion and Limitations

We observe only marginal gains by incorporating identity, CLS, and CLIP losses in our experiments—even after trying different metrics such as Cosine, L2, and Contrastive. These losses seem overshadowed by the adversarial loss, given that our discriminator (powered by BiomedCLIP's ViT and MSE loss) can detect fake images early on, effectively functioning as a strong regularizer. In single-domain settings, the broad, generalized embeddings (e.g., from OpenCLIP) can further destabilize this adversarial training, misaligning with the narrower data distribution and producing noise-like outputs. As a result, CLIP guidance—prone to providing inaccurate translation directions (Sauer et al., 2023; Kwon and Ye, 2022)—loses additional effectiveness.

Moreover, because BiomedCLIP serves as our multi-modal foundation model, our approach inherits its contrastive pre-training strategy, which emphasizes semantically meaningful features at the expense of finer image details. We also found our method to be sensitive to caption choices for the target domain, but experimenting with diverse captioning styles did not yield improvements. Consequently, we adopt BiomedCLIP's simplest templates (e.g., "This MRI is XX-weighted," "This is pelvic MRI," or "This is pelvic CT"). We leave further exploration of this aspect to future work.

## 6. Conclusion

We propose an unsupervised multi-modal image translation framework employing a generative adversarial network which is empowered with a pre-trained vision-language model. Our framework improves upon the cycle-consistent translation models while enhancing the multi-domain translation performance with a reduced computational budget.

## Acknowledgments

This work was supported by the NSF CAREER Award under Grant 2045489.

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

## Appendix A. Related Works

**Cycle-consistent image translation.** Zhu et al. revolutionized the field of unsupervised image translation with their proposal of CycleGAN (Zhu et al., 2017). Yi et al. proposed DualGAN (Yi et al., 2017) which is a concurrent work with CycleGAN offering the same cycle-consistency loss. Various studies followed the cycle-consistency constraint for more faithful translation in the unsupervised setting. Liu et al. proposed UNIT (Liu et al., 2017) for uni-modal translation where a shared latent space is assumed between source and target modalities. Huang et al. proposed MUNIT (Huang et al., 2018) where UNIT's assumption of shared latent space is divided into content and style for multi-domain translation. Lee et al. (Lee et al., 2018) introduced DRIT, which shares a similar approach to MUNIT by using disentangled content and attribute latents for multi-domain translation. Choi et al. proposed StarGANv1 (Choi et al., 2018) and StarGANv2 (Choi et al., 2020) where they utilized a separate style encoder network to generate distinct style codes to be used in generator for multi-domain translation. Perera et al. (Perera et al., 2018) proposed an alternative method where they utilize multi-domain input modalities with a latent-consistency loss. Kim et al. proposed U-GAT-IT (Kim et al., 2019) with an advanced generator equipped with adaptive layer instance normalization layers and attention. Torbunov et al. proposed UVCGan (Torbunov et al., 2023) employing a pre-trained vision transformer as generator in a cycle-consistent framework for improved translation performance.

**Text-guided image translation.**    Following the advancements in vision-language models ([Radford et al., 2021](#)) several text-guided unsupervised image translation methods proposed with or without cycle-consistency constraint. Park et al. proposed LANIT ([Park et al., 2023](#)) where they use CLIP to generate pseudo labels for unlabeled images with a similar approach in Starganv2. Gal et al. proposed StyleGAN-NADA ([Gal et al., 2022](#)) for CLIP driven adaptation of Stylegan2 generator ([Karras et al., 2020](#)). Patashnik et al. proposed StyleCLIP ([Patashnik et al., 2021](#)) where they invert source image to find its latent code for CLIP guided feature manipulation. Yu et al. proposed ([Yu et al., 2022](#)) a counterfactual image manipulation pipeline using CLIP.

**Diffusion-based image translation.**    More recently, building on the success of diffusion models in image generation, various unsupervised image translation methods utilizing diffusion-based backbones have been proposed. Zhao et al. proposed EGSDE ([Zhao et al., 2022](#)) where they utilize energy-guided translation between diversely trained diffusion models. Özbey et al. proposed SynDiff ([Özbey et al., 2023](#)), where they use multiple cycle-consistent diffusive and non-diffusive generators for improved translation performance. Kwon et al. proposed DiffuseIT ([Kwon and Ye, 2022](#)) and used pre-trained vision transformers as guidance in image manipulation. Tumanyan et al. ([Tumanyan et al., 2023](#)) offered a plug and play framework to adapt pre-trained text-to-image diffusion models in image translation. Zhan et al. proposed MedM2G ([Zhan et al., 2024](#)), where they proposed a unified multi-modal diffusive framework for text to image, image to text synthesis and image translation tasks. Our approach shares similarities with MedM2G ([Zhan et al., 2024](#)) in employing a multi-modal text-guided framework for image translation. However, our model is over an order of magnitude smaller, leveraging a feed-forward generative adversarial network architecture and enforcing cycle-consistency across translations. We also incorporate common loss terms with DiffuseIT ([Kwon and Ye, 2022](#)), utilizing CLS tokens from pre-trained vision transformers for semantically meaningful information extraction. Nonetheless, our approach differs in its use of cycle-consistency and the feed forward generative adversarial methodology adopted. We named our method in reference to the text-to-image generative adversarial model Galip ([Tao et al., 2023](#)). However, apart from the CLIP based feature extraction utilized for the Discriminator, our method does not share further similarities with Galip in terms of architecture or training methodology.

## Appendix B. Datasets and Competing Methods

### B.1. Datasets

1. **IXI**: Translation performance demonstrated in a single-coil brain MRI dataset from (http://brain-development.org/ixi-dataset/). $T_1$-, $T_2$- and PD-weighted acquisitions are considered. In IXI, 25 subjects are used for training, 5 for validation and 10 for testing.

2. **CT-MRI**: Translation performance demonstrated in pelvic $T_1$- and $T_2$-weighted MRI and CT data from ([Nyholm et al., 2018](#)). In CT-MRI dataset, 9 subjects are used for training, 1 for validation and 4 for testing.

### B.2. Competing Methods

1. **CycleGAN**: Cycle-consistent generative adversarial model is considered (Zhu et al., 2017). The Adam optimizer is utilized for training with an initial learning rate set at 0.0002, which linearly decreased to 0 after the $50th$ epoch. The training process spans a total of 100 epochs. Weights for adversarial, cycle, identity losses are selected as 1, 10, 0.5 respectively.

2. **U-GAT-IT**: An attention guided GAN model with adaptive layer-instance normalization designed for unsupervised image translation is considered (Kim et al., 2019). Adam optimizer is utilized for training with a learning rate of 0.0001. Training lasts for 100 epochs. Weights for adversarial, cycle, identity and CAM losses are selected as 1, 10, 10, and 1000 respectively.

3. **SynDiff**: A cycle-consistent diffusion-based image translation model is considered (Özbey et al., 2023). Adam optimizer is used for training with a learning rate of 0.0001. Training length is 50 epochs. The weights assigned to the cycle-consistency and adversarial loss terms are $\lambda_1^\phi, \lambda_1^\theta = 0.5$ and $\lambda_2^\phi, \lambda_2^\theta = 1$, respectively. The noise variance schedule is bounded between $\beta_{\min} = 0.1$ and $\beta_{\max} = 20$. Other diffusion related hyper-parameters are directly obtained from (Özbey et al., 2023).

4. **UNIT**: An unsupervised GAN model designed for unsupervised image translation is considered (Liu et al., 2017). Adam optimizer is utilized for training with a learning rate of 0.0001 for 100 epochs. Weights for adversarial, image, style, and content reconstruction losses are selected as 1, 10, 1, 1 respectively.

5. **EGSDE**: A diffusion based unpaired image translation model is considered (Zhao et al., 2022). Seperate DDPM models are trained for each translation domain to be utilized in EGSDE model. 500,000 diffusion steps are used for training of the DDPMs and T is selected as 150 to maintain source structure, and cross-validated weight parameters $\lambda_s$ and $\lambda_i$ are selected as $1 \times 10^{-7}$ and 10.

