# OpenReview forum: "I2I-Galip: Unsupervised Medical Image Translation Using Generative Adversarial CLIP"
_MIDL.io/2025/Conference — MIDL 2025 Poster_

### Official Review · Reviewer_Socs · 2025-02-13

**Confidence:** 5
**Preliminary Rating:** 4
**Recommendation:** Poster
**Final Rating:** 4

**Summary:**

The author introduce a method for unpaired image-to-image translation which leverages BiomedCLIP and adversarial learning, via its text encoder to which conditions Unet generator (based on a lightweight LDM architecture to handle the text conditioning), and it's image encoder as the base for a discriminator/critic. Comparing to a range of image-to-image translation models, they evaluate their method for multi-domain MRI<->MRI and single domain CT<->MRI translation via PSNR and SSIM (metrics that they can use due to a paired test set with ground truth output domain images). Their method achieves overall promising performance compared to existing approaches, while seemingly being lightweight.

**Strengths:**

1. The paper is overall well-written and well-structured, with figures which our illustrative of the method and its performance.
2. Only requiring a single network for the case of multi-domain translation is a notable benefit, as opposed to architectures like CycleGAN which directly scale with the number of domains considered.
3. The experimental evaluation is overall suitable for MIDL, in terms of: (a) The number/variety and difficulty of datasets/translation tasks, (b) the metrics used (no usage of FID or other perceptual metrics is OK given their established limitations for medical image translation as shown at MIDL 2024 (https://openreview.net/forum?id=acNu3glOTh), and (c) the quality and quantity of baseline/comparison translation models (although a few more notable models could be added if possible-- see weaknesses below).
4. Main experimental results: (a) for multi-domain MRI<->MRI translation: your method typically outperforms almost all competing methods, and comes close to the performance of a much bulkier model, SynDiff. (b) for single domain CT<->MRI translation: your method outperforms the baselines in general, although the overall performance gain is relatively small.
5. The ablation studies are appreciated, and helpful to understand the different importances of the model's many loss terms.

**Weaknesses:**

Major:

The primary limitation which must be considered is the performance of the proposed method when compared to other competing models. For multi-domain MRI<->MRI translation, there is not a convincing improvement over the baselines, particularly SynDiff. Your model(s) often perform second-best, but SynDiff usually is best. However SynDiff is definitely a heavy weight (diffusion-based) model which has seemingly higher compute demands and longer inference time (as you discuss in the paper). Therefore, your method would benefit from quantitatively (not just qualitatively) showing that the decreased performance compared to SynDiff is offset by improvements in inference (and training) time and compute -- please see my specific request for this mentioned below under Minor Weaknesses. For single domain CT<->MRI translation: your method outperforms the baselines in general, although the overall performance gain is relatively small. However, it should be noted that SynDiff actually performed notably worse in these experiments. hurting SynDiff when comparing it to your approach.

Minor (not ordered by importance):

1. The table 2 results are in the appendix, yet these are some the main results; these should be in the main text. Similarly, the ablation study section is a key component of the paper, yet the table with these results are in the appendix. The paper should stand on its own without having to refer to the appendix, so please include these in the main text.
2. Overall, the method is compared to baseline translation models which are strong in their quality and quantity. However, there are still a few important methods missing which the paper would benefit from comparing to them as well if possible. Their codebases are all pretty easy to use, so it should be fairly straightforward to evaluate (at least) one of these: (a) UNSB (ICLR 2024) (https://github.com/cyclomon/UNSB) (b) CUT (ECCV 2020) https://github.com/taesungp/contrastive-unpaired-translation and (c) GcGAN (CVPR 2019) https://github.com/hufu6371/GcGAN.
3. The technical/machine learning novelty is somewhat limited, but the usage of BiomedCLIP's encoders to drive adversarial learning of translation is clever and novel.
4. The authors use multiple discriminator "heads" given by outputs of different layers of BiomedCLIP's image encoder. However this makes the training/loss more complex, and could even make the model more bespoke to the tested datasets. It would be very helpful to understand the tradeoff of model complexity and performance gain derived from including multiple heads (as opposed to just a single encoder layers' feature space). Could the authors perform ablation studies accordingly with single vs. multiple heads?
5. A stated strength of the method is that it is lightweight (the generator has 13 M parameters), but this claim would be better supported and contextualized if you provided inference time and parameter count values for (A) your model, (B) an older yet still strong model like CycleGAN, and ( C) a recent, powerful, yet large model like SynDiff. Similarly, training time/compute estimates (per epoch) would be quite useful, if possible.
6. Math notation needs to be consistent, e.g. in 2.1, A, B, G, F are sometimes in math mode (italicized) and sometimes as normal text.
7. There a few minor typos and grammatical issues that the paper should be proofread for. E.g., "discriminator network is using" in lines 2-3 of 3.1 should be "discriminator network uses".

**Detailed Comments:**

My main suggestion is, as mentioned above, to quantitatively and clearly demonstrate that your model's improvements in inference/training time and compute offsets the slight performance drops which it has compared to SynDiff for multi-domain translation, which is stated as being a much larger/heavy-weight model (due to it being diffusion-based). While this seems to be the case qualitatively, the paper needs direct evidence of it. While the authors show that SynDiff is notably worse for single-image translation, CycleGAN is close to your method in performance while being older and seemingly lightweight, so again, it would be helpful to have direct quantified evidence for the tradeoff of how lightweight your model is compared to CycleGAN, with performance. The various minor weaknesses that I mentioned are not as important, but it would still be appreciated if you could address them, e.g. particularly comparing to UNSB, CUT, and/or GcGAN.

**Justification Of The Final Rating:**

Overall, I feel satisfied that the authors addressed my concerns. While I don't think your model should fully be labeled as "lightweight" given how close in training time and memory usage it is to SynDiff, yet still much larger than CycleGAN, the noticeable improvements in inference speed while still having good performance compared to SynDiff is commendable. However, the performance or computation gains over SynDiff are not so large as to boost this to a strong accept, so I maintain my rating of weak accept.

**Justification Of The Preliminary Rating:**

The paper is fairly strong overall, and presents a multi-domain image-to-image translation model which is more lightweight than related approaches, yet achieves fairly strong performance. As mentioned, properly quantifying how lightweight the model is compared to other related methods would greatly benefit the paper, so that practitioners know that it would be worth using this method over existing older lightweight, yet still fairly performant models like CycleGAN, or newer heavy-weight diffusion-based models like SynDiff which outperforms the proposed method for their evaluated multi-domain translation tasks, although being notably worse for single-domain translation.

**Questions To Address In The Rebuttal:**

As mentioned, the main thing that the authors can address which would improve my score is to quantifiably demonstrate that their model is lightweight (in compute time and parameter count) for inference/training compared to other approaches. While this is reasonably the case based off of the submitted manuscript and my knowledge, direct evidence would be helpful, to clearly place your method on the spectrums of performance and model complexity with respect to other approaches like SynDiff and CycleGAN.

One other small question thought: The loss has many terms (as is common for image-to-image translation models). Were there any difficulties in getting the model to converge? Did you use a vanilla GAN instead of something like a Wasserstein GAN with gradient penalty (https://arxiv.org/abs/1704.00028)? (the latter typically converges more reliably).

**Special Issue:**

No

---

> ### Author Response · Authors · 2025-03-06
> **Answer to questions raised by the reviewer Socs**
>
> Dear Reviewer,
>
> Responses to Questions & Major Weaknesses:
>
> 1) Thank you for your detailed suggestions and comments. We have provided the parameter counts for each competing method, including separate counts for generator-discriminator pairs and their total sum, in Table 4. Additionally, we report the inference time, memory usage, and training time requirements for CycleGAN, SynDiff, and I2I-Galip evaluated on a NVIDIA A5000 GPU with 24GB memory:
>
> CycleGAN:
> - Training time: approximately 4.5 hours for 100 epochs
> - Memory usage:
>   - 9,928 MiB total (batch size = 4)
>   - 3,128 MiB per batch (batch size = 1)
> - Inference speed: 0.00353 seconds per image
>
> I2I-GALIP:
> - Training time: approximately 27 hours for 100 epochs
> - Memory usage:
>   - 21,616 MiB total (batch size = 4)
>   - 7,074 MiB per batch (batch size = 1)
> - Inference speed:
>   - Text encoding: 0.01290 seconds per text prompt
>   - Image generation: 0.03593 seconds per image
>   - Total processing time: 0.04883 seconds per image
>
> SynDiff:
> - Training time: approximately 35 hours for 50 epochs
> - Memory usage:
>   - 26,254 MiB total (batch size = 4, evaluated on an A6000 GPU with 48GB memory due to memory overflow)
>   - 9,638 MiB per batch (batch size = 1)
> - Inference speed: 0.15792 seconds per image
>
> 2) To enhance stability in our pipeline, we utilize the least squares GAN loss [1], which has shown better stabilization compared to the vanilla GAN loss, following CycleGAN. This is discussed in Page 3 under Adversarial Loss.
>
> Responses to Minor Weaknesses:
>
> (1) The initial page limit was 8 pages, which required us to place some key results in the appendix. However, for the rebuttal, we will move as many results as possible, including Table 2 and the ablation study results, into the main text to ensure the paper stands on its own without reliance on the appendix.
>
> (2, 4) If time permits, we will include numerical results for at least one of the mentioned methods, as well as the suggested ablation study, in our rebuttal. We appreciate the provided references and recognize the importance of comparing our approach to a broader set of baselines.
>
> (6, 7) We fixed these issues for the rebuttal, ensuring consistency in mathematical notation, and we corrected minor typos and grammatical errors. We are going to update manuscript once we get the additional results mentioned above.
>
> We appreciate your feedback and look forward to your further insights.
>
> References:
>
> [1] Xudong Mao, Qing Li, Haoran Xie, Raymond YK Lau, Zhen Wang, and Stephen Paul Smolley. Least Squares Generative Adversarial Networks. In Proceedings of the IEEE International Conference on Computer Vision, pages 2794–2802, 2017.

---

> > ### Comment · Reviewer_Socs · 2025-03-06
> > **Thank you for these additional results**
> >
> > I thank the authors for providing thorough statistics on training time, memory usage and inference speed for CycleGAN, their model and SynDiff. Their model is understandably slower than the older CycleGAN, but the boosts in performance make it worth it, in my opinion. Compared to SynDiff, your model has slight improvements in training time and memory usage, and more noticeable improvements in inference speed, while being, on average, roughly competitive in terms of performance. Also, please make sure to add these results/numbers somewhere in your paper upon camera ready if it is accepted.
> >
> > Overall, I feel satisfied that the authors addressed my concerns. While I don't think your model should fully be labeled as "lightweight" given how close in training time and memory usage it is to SynDiff, yet still much larger than CycleGAN, the noticeable improvements in inference speed while still having good performance compared to SynDiff is commendable. However, the performance or computation gains over SynDiff are not so large as to boost this to a strong accept, so I maintain my rating of weak accept.

---

### Official Review · Reviewer_4rFJ · 2025-02-22

**Confidence:** 3
**Preliminary Rating:** 4
**Recommendation:** Poster
**Final Rating:** 4

**Summary:**

This paper proposes I2I-Galip, a cycle-consistent GAN framework to address unpaired multi-domain medical image translation. The authors use BiomedCLIP’s frozen encoders to eliminate the need for separate generator-discriminator pairs per domain and achieve efficient multi-domain translation with a lightweight U-Net generator.

**Strengths:**

1. The paper is well-written and easy to follow.
2. The integration of BiomedCLIP into a GAN framework is a novel and effective approach for multi-domain medical image translation.
3. The lightweight design and fast inference speed are more computationally efficient.

**Weaknesses:**

1. The evaluation metric only uses PSNR and SSIM, which may not fully capture downstream task performance.
2. It would be good to have more flexible prompts for BiomedCLIP to improve the framework’s adaptability.

**Detailed Comments:**

The paper's methodology and findings seem to be valid and align with similar work.

**Justification Of The Final Rating:**

The authors provide reasonable explanations for using fixed descriptive prompts. The justification for PSNR and SSIM with qualitative examples is adequate. I encourage future work to explore more flexible text prompting systems. I would like to maintain the rating of weak accept.

**Justification Of The Preliminary Rating:**

Overall, this paper presents an interesting approach to medical image translation by leveraging BiomedCLIP to achieve efficient and high-quality results. The lightweight design and improved inference speed are significant practical contributions in medical image translation. However, the lack of clinical validation and reliance on fixed text prompts limit its impact.

**Questions To Address In The Rebuttal:**

1. Can the method work with free-form text (e.g. doctor’s notes) instead of fixed prompts?
2. How do training time and memory compare to other models like CycleGAN?

**Special Issue:**

No

---

> ### Author Response · Authors · 2025-03-06
> **Answer to questions raised by the reviewer 4rFJ**
>
> Dear reviewer,
>
> Responses to Questions:
>
> Thank you for your insightful question. Our current approach utilizes fixed descriptive prompts to specify the target contrast in multi-domain image translation. While free-form text, such as doctor’s notes, could offer richer contextual information, directly incorporating it presents challenges due to BiomedCLIP’s limited context length (256 tokens). Additionally, the inherent variability of free-form text may introduce inconsistencies that could impact model stability. Nevertheless, developing strategies to effectively integrate structured information from free-form text while preserving robustness remains an intriguing direction for future research.
>
> We provide a detailed breakdown of the training time, memory usage, and inference speed for CycleGAN, SynDiff, and I2I-Galip, evaluated on a single NVIDIA A5000 GPU with 24GB memory.
>
> CycleGAN:
> - Training time: approximately 4.5 hours for 100 epochs
> - Memory usage:
>   - 9,928 MiB total (batch size = 4)
>   - 3,128 MiB per batch (batch size = 1)
> - Inference speed: 0.00353 seconds per image
>
> I2I-GALIP:
> - Training time: approximately 27 hours for 100 epochs
> - Memory usage:
>   - 21,616 MiB total (batch size = 4)
>   - 7,074 MiB per batch (batch size = 1)
> - Inference speed:
>   - Text encoding: 0.01290 seconds per text prompt
>   - Image generation: 0.03593 seconds per image
>   - Total processing time: 0.04883 seconds per image
>
> SynDiff:
> - Training time: approximately 35 hours for 50 epochs
> - Memory usage:
>   - 26,254 MiB total (batch size = 4, evaluated on an A6000 GPU with 48GB memory as the A5000 experienced memory overflow).
>   - 9,638 MiB (batch size = 1)
> - Inference speed: 0.15792 seconds per image
>
> Responses to Weaknesses:
>
> Thank you for your comment. While we used PSNR and SSIM as quantitative evaluation metrics, we also provided qualitative examples to demonstrate the effectiveness of our method in preserving local details. These visual results offer additional insights into how well our approach maintains structural and textural consistency across different translations, complementing the numerical metrics. We believe that qualitative assessment is crucial for capturing aspects of image quality that traditional metrics may not fully reflect, especially in medical imaging tasks where fine-detail preservation is essential.
>
> We appreciate your feedback and look forward to your further insights.

---

> > ### Comment · Reviewer_4rFJ · 2025-03-14
> >
> > The authors provide reasonable explanations for using fixed descriptive prompts. The justification for PSNR and SSIM with qualitative examples is adequate. I encourage future work to explore more flexible text prompting systems. I would like to maintain the rating of weak accept.

---

### Official Review · Reviewer_HMrR · 2025-02-26

**Confidence:** 4
**Preliminary Rating:** 4
**Recommendation:** Poster

**Summary:**

By integrating BiomedCLIP, it improves translation fidelity while requiring fewer parameters. However, its reliance on text descriptions and BiomedCLIP’s pre-training strategy introduces some limitations in terms of fine-detail preservation and generalization to single-domain tasks.

**Strengths:**

Using BiomedCLIP, it improves translation quality, reduces artifacts, and enhances inference speed. Its cross-attention transformers enable text-guided control, preserving anatomical features while outperforming diffusion and cycle-consistent models in PSNR, SSIM, and computational efficiency across MRI and CT modalities.

**Weaknesses:**

It is sensitive to caption choices, impacting translation accuracy. Additionally, BiomedCLIP’s contrastive pre-training prioritizes semantic features over fine details, potentially limiting fidelity. Single-domain settings may destabilize adversarial training.

**Detailed Comments:**

By integrating BiomedCLIP, it improves translation fidelity while requiring fewer parameters. However, its reliance on text descriptions and BiomedCLIP’s pre-training strategy introduces some limitations in terms of fine-detail preservation and generalization to single-domain tasks.

**Justification Of The Preliminary Rating:**

While the paper appears to contribute meaningfully to the field, there are concerns regarding the impact of specific loss functions, model sensitivity, and comparisons with other vision-language models. If these issues are significant enough to affect confidence in the results, it leans towards weak accept.

**Questions To Address In The Rebuttal:**

They should address the model's sensitivity to caption variations and explore alternative text encoding methods. A comparison with other vision-language models, like OpenCLIP, would help assess BiomedCLIP’s effectiveness. Additionally, explaining why adversarial loss dominates in single-domain settings and proposing solutions for stability would be valuable. Given the emphasis on semantics over fine details, techniques for preserving image structure should be considered. Lastly, discussing the trade-off between computational efficiency and image fidelity could provide insights into potential model improvements.

**Special Issue:**

No

---

> ### Author Response · Authors · 2025-03-06
> **Answer to questions raised by the reviewer HMrR**
>
> Dear reviewer,
>
> Responses to Questions:
>
> Thank you for your valuable suggestions regarding alternative text encoding methods. As demonstrated in our ablation study, BiomedCLIP significantly outperforms OpenCLIP in our experiments (see Table 3). We acknowledge the importance of exploring diverse approaches; however, our study is constrained to medical foundation model backbones.  Given that BiomedCLIP is one of the most widely recognized foundation models in the medical domain, we selected it as the most suitable choice for our work.
>
> Responses to Weaknesses:
>
>   Although we experimented with a diverse range of captions, we did not observe significant differences in outcomes, as adversarial training and cycle-consistency primarily drive the translation process. While BiomedCLIP’s contrastive pre-training emphasizes semantic representations, this can limit its ability to capture the fine-grained details crucial for medical imaging. To mitigate this, our model employs multiple small discriminator heads, each specialized for a specific target domain, ensuring the preservation of both structural and textural information. Additionally, cycle-consistency loss helps refine local details and improve translation fidelity.
>
> Future work could focus on fine-tuning BiomedCLIP to enhance its feature extraction capabilities, making it more sensitive to local details while retaining its strong semantic understanding. A hybrid approach that combines contrastive learning with pixel-level objectives may further improve the balance between global semantics and fine-detail preservation, leading to more precise and contextually accurate image translations.
>
> We appreciate your feedback and look forward to your further insights.

---

### Official Review · Reviewer_3Qnc · 2025-02-26

**Confidence:** 4
**Preliminary Rating:** 2
**Final Rating:** 3

**Summary:**

This paper presents a I2I translation framework where the authors use pretrained multi-model foundation models and a lightweight generator.

**Strengths:**

The authors present a unsupervised adversarial framework for  language-driven multi-domain medical image translation. The qualitative results look promising. Clear implementation details and configurations.

**Weaknesses:**

Limited innovations in the overall methodology, which is simply combine all prior arts in the proposed method with no significant improvements.

The SynthDiff performance tends to be better in majority cases in Table 1, which does not suggest the proposed method has superior performance than others.

Lacks of significant testing when comparing with other methods.

Lacks of hyperparameter tuning since there are a lot of hyperparameters involved in the loss function.

Some essential quantitative results, ablation study, computation analysis are put in the appendix, which is not conventional.

**Detailed Comments:**

Please refer to the weakness for improvements.

**Justification Of The Final Rating:**

The authors answered majority of the questions well.

However, the significant testing (p-values) results still missing when comparing with other competing methods.

Overall, I updated my rating based on the responses.

**Justification Of The Preliminary Rating:**

This paper is not novel enough for the overall methodology development. As mentioned in the weakness part, there are a lot of issues needed to be addressed first.  There are a lot of loss functions in the method, which suggest the proposed model could be challenge to tune, and there were no hyperparameter finding process in this paper.

**Questions To Address In The Rebuttal:**

Please list out the innovation parts in the proposed method itself.

---

> ### Author Response · Authors · 2025-03-06
> **Answer to questions raised by the reviewer 3Qnc**
>
> Dear Reviewer,
>
> Responses to Questions:
>
> Thank you for your comments. Below, we summarize the key innovations of our proposed model:
>
> 1) Our approach uniquely leverages the extensive radiological expertise embedded in BiomedCLIP to enhance medical image translation. By employing a unified, lightweight network architecture, we address multi-domain image translation tasks across various medical imaging modalities. This design expands the applicability of BiomedCLIP-like foundation models beyond traditional tasks such as question answering or medical report generation, integrating deep radiological insights into a broader range of applications.
>
> 2) We utilize BiomedCLIP’s Vision Transformer as our foundational feature extractor—an approach applied in text-to-image scenarios like StyleGAN-T [1] but not previously explored for medical image translation task. Instead of deploying separate, full-scale discriminator networks, we introduce multiple small discriminator heads, each specialized for a specific target domain. The robust feature extraction capabilities of BiomedCLIP allow these compact heads to capture domain-specific details effectively. To validate our methodology, we compared it against OpenCLIP trained on the LAION dataset, demonstrating that our approach effectively utilizes pre-trained radiological knowledge during feature extraction for the discriminator heads and CLIP-guided losses.
>
> Responses to Weaknesses:
>
> 1) We have outlined our core innovations above, primarily focusing on designing a novel framework that incorporates medical pre-training in a non-downstream task: unsupervised image translation.
>
> 2) SynDiff incurs a substantial computational cost in terms of both parameter count (see Table 4) and inference time due to its iterative diffusion process. Given SynDiff's multi-stage training process, where it first trains a non-diffusive module followed by a diffusive module, along with its iterative diffusion process during inference, superior results are expected in certain cases, particularly when this sequential training and iterative refinement contribute to more precise image generation. We have further detailed its inference time, memory, and training requirements below, evaluated on a NVIDIA RTX A5000 GPU with 24GB memory.
>
> 3) We conducted experiments on two publicly available datasets, covering eight distinct translation tasks across various imaging modalities and anatomical regions, including brain MRI and pelvic CT. Please let us know if additional tests are needed to further enhance the evaluation.
>
> 4) Hyperparameter ablation studies, presented in Table 3, illustrate the contributions of different loss components to the overall performance of our model.
>
> 5) The initial page limit was 8 pages. For the final version, we will incorporate additional results into the main text.
>
> I2I-Galip:
> - Training time: approximately 27 hours for 100 epochs
> - Memory usage:
>   - 21,616 MiB total (batch size = 4)
>   - 7,074 MiB per batch (batch size = 1)
> - Inference speed:
>   - Text encoding: 0.01290 seconds per text prompt
>   - Image generation: 0.03593 seconds per image
>   - Total processing time: 0.04883 seconds per image
>
> SynDiff:
> - Training time: approximately 35 hours for 50 epochs
> - Memory usage:
>   - 26,254 MiB total (batch size = 4, evaluated on an A6000 GPU with 48GB memory as the A5000 experienced memory overflow).
>   - 9,638 MiB per batch (batch size = 1)
> - Inference speed: 0.15792 seconds per image
>
> We appreciate your feedback and look forward to your further insights.
>
> References:
> [1] Sauer, Axel, et al. "Stylegan-t: Unlocking the power of gans for fast large-scale text-to-image synthesis." International conference on machine learning. PMLR, 2023.

---

### Author Rebuttal · Authors · 2025-03-08

**Rebuttal:**

We sincerely thank the reviewers for their valuable comments and suggestions. In this rebuttal, we have moved significant results from the appendix to the main text and introduced a new computational cost table. We have also corrected grammatical errors and improved consistency in notation. The changes in the text are highlighted in dark purple.

**Supporting Material:**

/attachment/e8ea6a3dd59d407edd66edcb71e91ca0927471d1.pdf

---

### Meta-Review · Area_Chair_59gj · 2025-03-19

**Recommendation:** Accept (Poster)
**Confidence:** 3

**Metareview:**

All reviewers found the proposed method to be novel and the results promising.